# Seek and you may (not) find: A multi-institutional analysis of where research data are shared

Lisa R. Johnston[1], Alicia Hofelich Mohr[2], Joel Herndon[3‡*], Shawna Taylor[4‡], Jake R. Carlson[5‡], Lizhao Ge[6‡], Jennifer Moore[7‡], Jonathan Petters[8‡], Wendy Kozlowski[9‡], Cynthia Hudson Vitale[10‡]

1 Data, Academic Planning & Institutional Research, University of Wisconsin-Madison, Madison, Wisconsin, United States of America, 2 Liberal Arts Technologies and Innovation Services, University of Minnesota, Minneapolis, Minnesota, United States of America, 3 Center for Data and Visualization Sciences, Duke University Libraries, Duke University, Durham, North Carolina, United States of America, 4 Association of Research Libraries, Washington, D.C., United States of America, 5 University at Buffalo Libraries, University at Buffalo, Buffalo, New York, United States of America, 6 Milken Institute School of Public Health, George Washington University, Washington, D.C., United States of America, 7 University Libraries, Washington University in St. Louis, St. Louis, Missouri, United States of America, 8 Data Services, University Libraries, Virginia Tech, Blacksburg, Virginia, United States of America, 9 Research Data and Open Scholarship, Cornell University Library, Cornell University, Ithaca, New York, United States of America, 10 Association of Research Libraries, Washington, DC., United States of America

☯ These authors contributed equally to this work.
‡ JH, ST, JRC, LG, JM, JP, WK and CHV also contributed equally to this work.
* joel.herndon@duke.edu

**Data Availability Statement:** All code and associated data files are available from Zenodo general purpose open respository (https://zenodo.org/records/8368205).

## Abstract

Research data sharing has become an expected component of scientific research and scholarly publishing practice over the last few decades, due in part to requirements for federally funded research. As part of a larger effort to better understand the workflows and costs of public access to research data, this project conducted a high-level analysis of where academic research data is most frequently shared. To do this, we leveraged the DataCite and Crossref application programming interfaces (APIs) in search of Publisher field elements demonstrating which data repositories were utilized by researchers from six academic research institutions between 2012–2022. In addition, we also ran a preliminary analysis of the quality of the metadata associated with these published datasets, comparing the extent to which information was missing from metadata fields deemed important for public access to research data. Results show that the top 10 publishers accounted for 89.0% to 99.8% of the datasets connected with the institutions in our study. Known data repositories, including institutional data repositories hosted by those institutions, were initially lacking from our sample due to varying metadata standards and practices. We conclude that the metadata quality landscape for published research datasets is uneven; key information, such as author affiliation, is often incomplete or missing from source data repositories and aggregators. To enhance the findability, interoperability, accessibility, and reusability (FAIRness) of research data, we provide a set of concrete recommendations that repositories and data authors can take to improve scholarly metadata associated with shared datasets.

**Funding:** Funding for this research and the Realities of Academic Data Sharing (RADS) Initiative was provided by the National Science Foundation (NSF), award #2135874, EAGER grant: Completing the Lifecycle: Developing Evidence Based Models of Research Data Sharing. The funders had no role in study design, data collection and analysis, decision to publish, or preparation of the manuscript.

**Competing interests:** Institutions involved in this study maintain paid memberships in either the CrossRef or DataCite data sharing services. This does not alter our adherence to PLOS ONE policies on sharing data and materials.

## Introduction

The number of available public access repositories for data and code has exploded over the last ten years. The Registry of Research Data Repositories (re3data.org), which launched in 2012 to index research data repositories available to researchers, tallied 400 research data repositories in July 2013 [1]. This number grew to 1,821 repositories in 2015 [2] and at the time of writing in 2023, the registry included 3,148 registered repositories [3]. This explosive growth has compounded the difficulty of answering the question: where are researchers at US academic institutions sharing their data? Additionally, the last decade has seen the growth of research information management systems that aggregate information about faculty publications or peer collaboration networks [4]. However, the large-scale discovery and aggregation of information about research data has only recently become feasible given the growth in data repositories and the adoption of digital object identifiers (DOIs) for datasets, which is thanks to several open data metrics efforts [5, 6]. As federal funders build and implement requirements for data and other research products to be shared and made publicly accessible, institutions can start asking how well their teaching, research, and outreach mission is met by analyzing research data shared by their researchers.

In theory, finding where researchers share their data should be relatively easy using recent advancements in research information management infrastructure. DOIs minted through the use of services such as DataCite and Crossref provide a persistent identifier (PID) and capture metadata about a publicly accessible dataset, including information about the author or creator. However, while tools such as DataCite have existed for many years, in practice, search and discovery across the metadata based on author affiliation is challenging. For example, DataCite only added the "affiliation" attribute to their metadata schema in version 3.1 [7] and did not *require* affiliation information for authors until several years later. A real-life example of this limitation is demonstrated in Montana State University Library's Dataset Search project which attempted to harvest metadata of datasets based on the affiliation field. After some trial and error, this project team instead based their queries on known author names for all faculty within the organization with disambiguation based on author department, rather than relying on the limited sample yielded via affiliation search [8]. Despite the growth in publishing data as a valuable output of the research process, it is still very difficult to identify where specific datasets have been published and to trace the data back to the originating institution.

Searching for where data are shared is further complicated by the fact that research data are shared in a multitude of ways including on personal author websites, standalone web pages, author-maintained databases, or as supplemental files to the research article. Recent research has indicated that the majority of data availability statements in PLOS journals state that research data will be shared in the paper or a combination of in the paper and the supplemental files, therefore making unique, stand-alone citation impractical, if not impossible [9].

When data are shared as published standalone artifacts, author affiliation information is often incomplete or uses short name forms or sub-units of the institution (e.g., Harvard Med School). Author affiliation is not static and may change several times over a scholar's career. In an effort to standardize the practice of authorship and affiliation information within metadata, registries such as ORCID and the Research Organization Registry (ROR) aim to disambiguate researcher names and organizational names, respectively. While incredibly useful, the development and use of ORCID and ROR is relatively recent, and adoption is inconsistent throughout the scholarly communications landscape. For example, RORs were not accommodated in the DataCite schema until version 4.3 [10, 11]. Thus, knowing where *all* research data are shared by *all* researchers within an institution is not easily known [12].

This presents a clear challenge for organizations who require accurate data about these metrics to inform investment or meet compliance requirements. For example, senior administrators may need to make informed decisions about investments in repository infrastructure and resources for staffing and services. Knowing the landscape of top research data publishers for their institution would help inform and shape those decisions. Institutions may also rely on data sharing metrics for compliance reporting around growing federal open access requirements, accountability in the case of research integrity investigations, and as metrics for academic benchmarking, particularly as data are recognized as distinct and important outputs of academic scholarship. To effectively use these metrics, it is critical to have an understanding of both the data sharing landscape as well as factors that can bias our understanding of this landscape. For example, if relying on large-scale metadata records for these answers, it is critical not only to understand how the metrics were gathered and assessed, but to understand the completeness and quality of the records. Differences in metadata completeness may lead to systematic omissions of important repositories in the data sharing landscape (for example, repositories with missing or inconsistent entries for affiliation, funder, or other fields would be absent from queries using those fields), while differences in metadata quality may critically affect the interpretation of the landscape (for example, repositories may differ in their use of "data" or "software" to describe a collection of data files underlying a study).

## Research questions

To further examine these challenges, the Realities of Data Sharing (RADS) project led by the Association of Research Libraries (ARL) was formed in partnership with the Data Curation Network and six academic research institutions, Cornell University (Cornell), Duke University (Duke), University of Michigan (Michigan), University of Minnesota (Minnesota), Virginia Tech, and Washington University in St Louis (WashU). All of these institutions have doctoral-granting programs and are classified as having very high research activity (Carnegie classification R1), with federal sponsored research expenditures ranging from $240 million to over $970 million [13]. The institutions range in size with total enrollments of 16,000 to 52,000 and are a mix of private (n = 3) and public (n = 3) universities. The aims of this project were to understand where academic researchers share their data, based on author affiliation and DOI assignment; to explore what motivates researchers' decision-making processes for sharing; and to determine the costs involved for data sharing to both the researcher and institution [14, 15].

In this paper, we present results from the first of these aims, striving to answer the following questions about data sharing at our six universities:

RQ1: How many datasets are being shared?

RQ2: Where are research data being shared?

RQ3: How complete are the metadata records associated with these datasets?

Our method and workflow may be useful to other academic institutions and higher education stakeholders who wish to deepen their knowledge of data sharing practices at their respective institutions. As data curators with experience in managing data repositories within our institutions, we further reflect on our methods and results in order to make recommendations to enhance data sharing metrics. As a result, these recommendations could potentially increase the ability for staff at both institutions and repositories to better evaluate, interpret, and support data sharing.

## Methods

To approach our research questions, we acquired metadata from sources that register Digital Object Identifiers (DOI) for datasets or code published in the United States and that also denote author affiliation. Given the exploratory nature of this work, our protocol was unfixed and we describe our work, including modifications and limitations of the process we implemented. Out of the eleven official DOI Registration Agencies worldwide, RADS researchers selected DataCite [16] and Crossref [17] data sources for this project, as they are the primary DOI service providers used in the US; both also specify the object type "dataset" (additionally DataCite includes "software"). There are inherent advantages and disadvantages to using DOI-specific minting services for sourcing our information about where researchers are sharing their data. Although it is increasingly becoming standard practice, a Digital Object Identifier (DOI) is just one *type* of the more generic persistent identifier (PID). Other forms, which lack the centralized registries DOIs afford, such as the Archival Resource Key (ARK), Handle System identifiers, the eponymously named PURL system, and other internal IDs, are still utilized by some data repositories and archives. Repositories push metadata about the datasets to the DOI service providers when a DOI is issued, providing a common and authoritative source of metadata. However, repositories differ in how much metadata is pushed into the DOI service providers, both due to initial structure and collection of the metadata in the repository, how easily metadata is crosswalked to from the repository to the DOI provider, the number of required fields submitted by the author, and the extent of curation provided by the repository.

DataCite also presents another challenge since it includes holdings from over 1,000 data repositories worldwide, but, it is unclear how comprehensively metadata from these repositories are transferred and integrated into the DataCite index. This includes whether optional yet valuable information, such as author affiliation, is included or if DataCite is being used to function primarily as an external microservice to mint a DOI with minimally required metadata, suggesting that authoritative metadata management is retained in home systems [18]. This is the case for many of the RADS institutional data repositories, where metadata schemas utilized in the underlying repository infrastructure do not align with those used by DOI providers, making complete integration of metadata a manual or highly custom task. As a result, institutions often submit only the bare minimum of metadata to DataCite to receive a DOI. One of our challenges, therefore, is to account for this lack of integration between known institutional repositories and the global research data sharing metadata infrastructure, and still discover as many institutionally affiliated datasets as possible.

### Data collection

To scope the project, we searched for datasets authored by at least one affiliate from the six RADS institutions with a DOI published between January 2012 and April 2022 (Crossref) or October 2022 (DataCite). Both DOI service providers were searched based on author affiliation metadata.

**DataCite**: DataCite metadata at the "Client" level were pulled using the rdatacite [19] package in R statistical software [20] by searching the University names in the DataCite *creator. affiliation* field. This field is defined in the schema as "The organizational or institutional affiliation of the creator" in the DataCite metadata schema [10]. Because we were interested in looking across data from the last ten years, we were unable to take advantage of ROR identifiers, as DataCite only implemented the support of RORs in the affiliation field as an optional field in 2019 [11]. Results were filtered to remove paper publications by faceting *resourceType-General* to "dataset" or "software" and the relevant institutional affiliations.

**Table 1. Institutional repositories included in sample.**

| Institution | Counts found by affiliation in initial search | Primary Institutional Repository for Data |
|---|---|---|
| Cornell University | 34 | Cornell eCommons, http://ecommons.cornell.edu/ |
| Duke University | 1 | Duke Research Data Repository, http://research.repository.duke.edu/ |
| University of Michigan | 1 | Michigan Deep Blue Data, https://deepblue.lib.umich.edu/data |
| University of Minnesota | 77 | Data Repository for University of Minnesota (DRUM), https://conservancy.umn.edu/drum |
| Virginia Tech | 113 | Virginia Tech Data Repository, https://data.lib.vt.edu/ |
| Washington University in St. Louis (Wash U)[a] | 0 | Washington University Data Repository, https://openscholarship.wustl.edu/data/ |

[a]The Washington University Data Repository was replatformed in 2023 to WashU Research Data: https://data.library.wustl.edu/. All data reported here was collected on the older platform. The number of results found for each IR in the original affiliation-based searches are included.

**Crossref**: Following recommendations of the Crossref API [21], metadata was pulled from the April 2022 Public Release file [22]. Records were selected from this file that had one of the RADS institutions in the element *affiliations.institution*, defined in the Crossref schema as "Container element for information about an institution or organization associated with an item" [23], and that had *created-dateparts* year of 2012 or newer, and *type* as "datasets", as CrossRef does not have software as an available type. Similar to DataCite, RORs in Crossref could not be leveraged as Crossref only began supporting RORs in 2021 [24].

At this point in our collection, we discovered that known datasets housed in our local institutional repositories were missing author affiliation information and therefore did not appear in either of the two searches (across all six institutions, DataCite included 226 datasets and Crossref included 0). Acknowledging that this introduces a divergence from our original search method, we recognized failing to adequately capture institutional repositories, which are dedicated only to authors affiliated with an institution, would create a critical gap in knowing where institutional data was published. Therefore, it was important to include our institutional data repositories and we added these as a third source in our queries (see Table 1).

**Institutional repositories**: To more fully capture metadata of datasets housed in our institutional repositories, which solely serve researchers affiliated with each of our institutions, we searched explicitly for each of the six RADS institutional repositories using the respective databases used to issue the repositories' DOIs. Five repositories used DataCite to issue DOIs; at the time of the queries, the sixth institution, Duke, used Crossref (and has since joined DataCite). For the five institutions that used DataCite, we queried the DataCite API by names of the institutional repositories in the *publisher* field and for Duke we used the Crossref API using the rcrossref R package [25] to retrieve all DOIs published using the Duke member prefixes. Institutional repository data were then filtered to include only datasets and software resource types, with DOIs published in 2012 or later. When multiple affiliated institutional repositories contained data for an institution (such as at Duke's "Digital Research Repository" and "Research Data Repository") these dataset counts were aggregated into one column. Metadata from datasets where the institution names were found in the *publisher* field were then reviewed and subsetted to include only the relevant institutional repositories from the returned publishers. Each team member reviewed the returned list of repositories affiliated with their institution and selected those considered repositories for data produced primarily by researchers at the institution. Some repositories based out of an institution appeared in our sample, but were

excluded if they did not function as the institutional repository. For example, MorphoSource, a repository based out of Duke University, was excluded in the institutional repository group because it is an NSF-funded repository of 3D natural history and publishes datasets independent of affiliation.

## Data analysis

DataCite and Crossref metadata records for datasets affiliated with our six institutions, including those queried by Institutional Repository name in the publisher field, were combined into a single dataset resulting in 181,812 records. The results were then pulled into R [20] and refined and analyzed within an R markdown file [26]. During the refinement process, described above, we first de-duplicated DOIs within the same institution, which removed the DOIs from the IRs initially found in the DataCite pull. Other stray duplicated DOIs, many of which were found in the ENCODE repository results, were also removed. This method did not remove DOIs that were co-authored across institutions, as each dataset would count for that institution's author. The refinement also addressed other incompatibilities with the records. For example, some repositories, such as Harvard's Dataverse and Qualitative Data Repository, assign DOIs at the file level rather than the study level. This resulted in some individual studies having multiple, and in some cases, thousands, of individual DOIs. Similarly, Zenodo often has many related DOIs for multiple figures within a study. Therefore, in order to compare study-to-study counts of data sharing, we reduced this dataset by collapsing datasets falling into the same container ID (e.g., one dataset for hundreds of files) for a final total of 143,633. Fig 1 illustrates the data analysis processes. The total records broken down by affiliation for each of the six institutions are displayed in Table 2.

Frequencies and descriptive statistics were used to assess the number of datasets and publishers for each institution. We focused on measuring metadata record completeness for fields recommended by the Office of Science and Technology Policy (OSTP) 2022 memo, "Ensuring Free, Immediate, and Equitable Access to Federally Funded Research," which states, "Such metadata should include at minimum: i) all author and co-author names, affiliations, and sources of funding, referencing digital persistent identifiers, as appropriate; ii) the date of publication; and, iii) a unique digital persistent identifier for the research output" [27]. We used completeness as a proxy for assessing quality, as quality is much more subjective and difficult to assess programmatically.

## Results

### How many datasets are being shared?

In total we observed 173 unique repositories housing 143,633 datasets affiliated with our six institutions. When assessed over time, we observed a steady increase in the diversity of unique repositories chosen by affiliates at each institution (see Fig 2).

### Where are research data being shared?

The data repositories with the most datasets unsurprisingly included some popular generalist repositories: Zenodo (which includes archived GitHub repositories), Dryad, Figshare, ICPSR, and Harvard Dataverse. Since all of our institutions host an institutional repository for data (and we specifically searched for our repositories), this category shot up to sixth place for our sample. After the top few publishers, the drop off is significant. The 20 top publishers ranked by total number of data DOIs are listed in Table 2 and the top 10 are shown by institution in Fig 3. The majority of datasets were found in a small number of publishers; 99% of the data

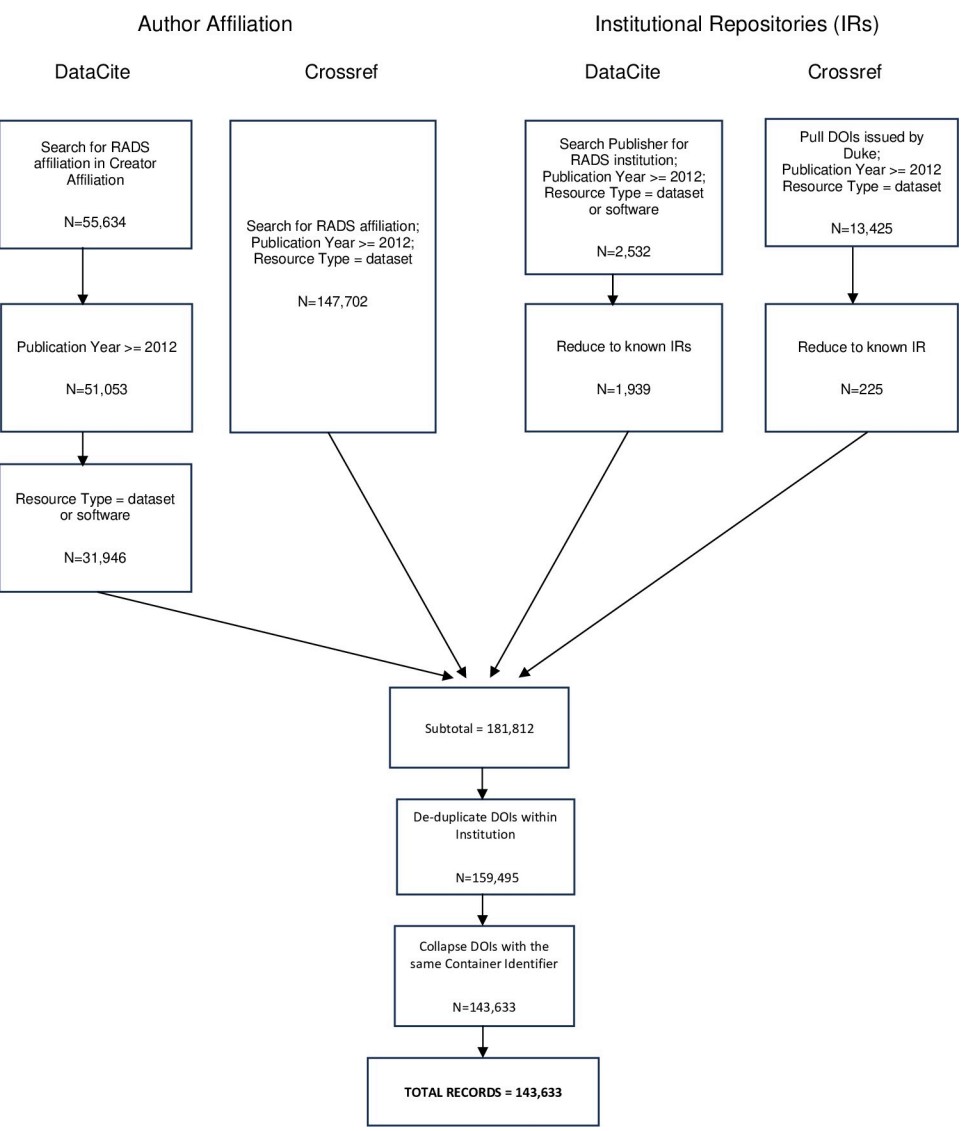

**Fig 1. Data collection and refinement method.** The Realities of Academic Data Sharing (RADS) flowchart for gathering and refining DOIs for data and software from DataCite and CrossRef sources.

DOIs were held within the top ten repositories, with this number varying between 89.0% and 99.8% for each individual institution.

However, we did observe two surprising publishers of datasets near the top of our list, both with DOIs generated by Crossref: ENCODE Data Coordination Center and Faculty Opinions Ltd. We examined these two repositories more closely to better understand the types of data each held. ENCODE publishes (many) genomic datasets and the majority were affiliated with the University of Michigan and Duke University [28]. Faculty Opinions Ltd (now H1 Connect) entries were reviews or commentary of life sciences and medical research [29], which did not include discernible datasets, despite being designated as "dataset" in the resource type metadata. We confirmed this via email correspondence with their managing director, Tiago Barros. The distribution of published datasets within each institution with and without these two publishers included are captured in Fig 4.

**Table 2. Total counts of datasets and software code, by author institutional affiliation.**

| Institution | Affiliated datasets | | | |
|---|---|---|---|---|
| | DataCite | Crossref | Institutional Data Repository | Total |
| Cornell University | 3,887 | 655 | 174 | 4,716 |
| Duke University | 2,370 | 2,969* | 225 | 5,564 |
| University of Michigan | 4,187 | 119,942* | 645 | 124,774 |
| University of Minnesota | 2,322 | 1,514 | 692 | 4,528 |
| Virginia Tech | 1,442 | 64 | 333 | 1,839 |
| Washington University in St. Louis (WashU) | 1,626 | 491 | 95 | 2,212 |
| Total by source | 15,834 | 125,635 | 2,164 | 143,633 |

*These results contain records from the ENCODE genomic sequence repository, which accounted for 82% of the total DOIs across institutions. The total number of non-ENCODE Crossref DOIs from Duke University was 2,443 and Michigan was 2,373.

Exploring these repositories by institution, as shown in Figs 3 and 4 and in Table 3, we see publisher patterns emerge both within and across institutions, which could be indicative of a local domain specialty. For example, Michigan, Duke, and Cornell are the only institutions with affiliated datasets in ENCODE, and several repositories arrive on our list exclusive to one institution, such as Neotoma Paleoecological Database with Minnesota, Virginia Tech Transportation Institute (VTTI) with Virginia Tech, and OBIS-SEAMAP with Duke.

Interestingly, as Table 3 shows, the top 10 publishers account for 89.0% to 99.8% of the datasets uncovered by our search with a large representation in Zenodo, Dryad, the local institutional repository, and Taylor & Francis. This large percentage could be indicative of several possibilities: these top 10 publishers could be more likely to include the affiliation metadata which formed the basis of our collection method; possibly, researchers are being driven to these top 10 publishers by some means (awareness, publisher recommendation, colleague recommendation, library recommendation, preference, etc.); finally, these top 10 publishers may be ingesting data that are produced in higher volumes in general, as we see with ENCODE and genomic datasets.

## How complete are the metadata records associated with these datasets?

In addition to analyzing where research data are shared, the research team did some preliminary analysis of the metadata associated with datasets in our sample to answer our research question about the completeness of the metadata records. As noted earlier, our data collection methods to retrieve DOIs from our RADS institutional repositories were modified based on the lack of affiliation metadata in the DataCite records. It is clear the lack of metadata completeness affects the answer to our question "where are the research data shared"; other data publishers with missing or incomplete affiliation metadata in the DOI record are completely missing from our results.

To further assess variation in the completeness of the metadata record within our results, we looked at the rates of presence versus absence of information in the DOI metadata record. Because the majority of data publishers in our sample issued DataCite DOIs, we examined specific metadata fields from the DataCite schema and captured whether the elements contained *any* information (which we will refer to as "complete" fields). DataCite records contain many metadata fields of varying applicability for every dataset, so we restricted our examination to metadata fields based on recommended information described in the 2022 OSTP public access memo [27], and are (starred in Table 4):

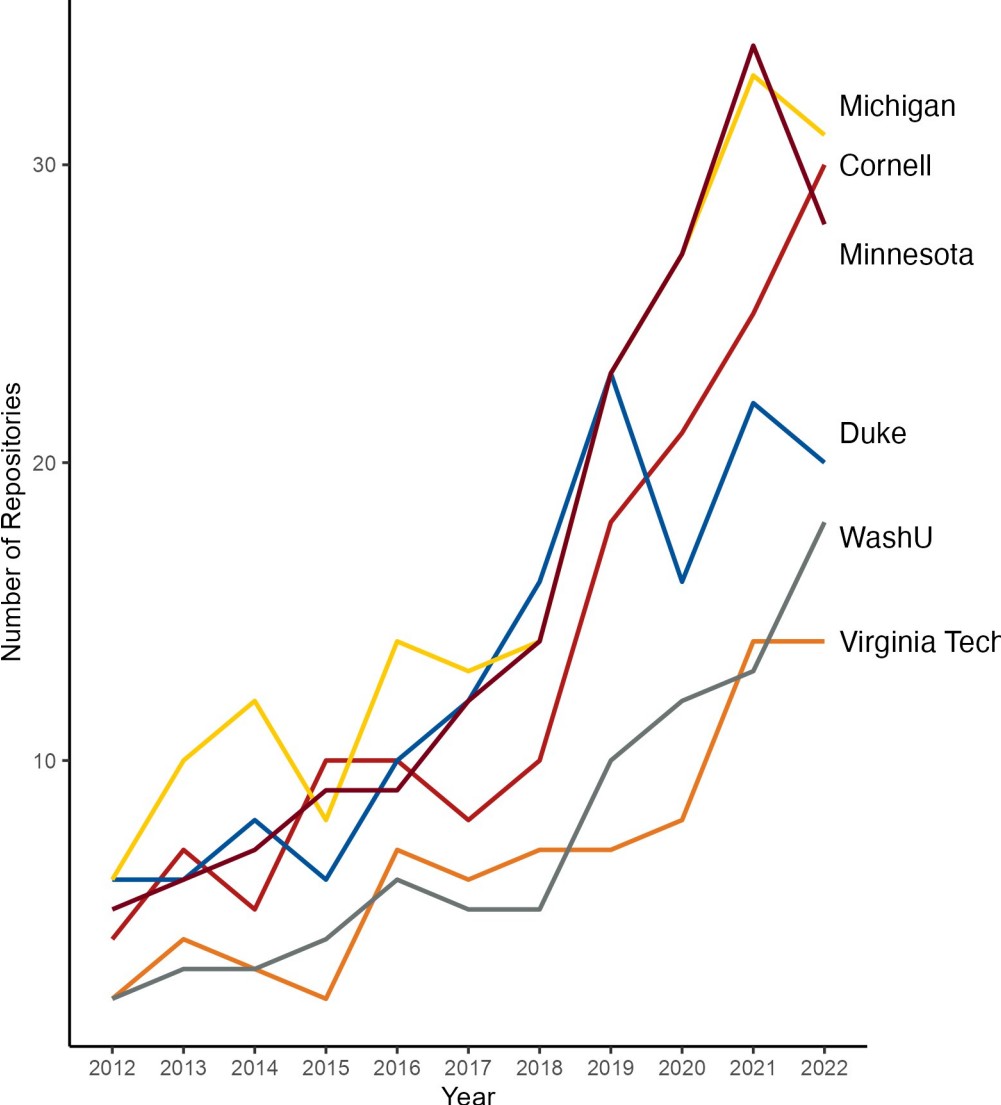

**Fig 2. Growth in number of repositories over time.** Count of repositories where data and software are shared by researchers from 2012 to 2022.

- Direct Object Identifier (DOI): The persistent unique identifier (PID) for the dataset minted by the repository

- Publication Year: Year in which data were published by the repository

- Creator Name: First and last name of creator

- Creator Affiliation: Affiliation of creator, such as a university name

- Related Identifiers: PIDs for related papers, datasets, or code often DOIs for related publications associated with the dataset

- Creator Name Identifier: PIDs associated with authors, often an Open Researcher and Contributor ID (ORCID)

- Funder Name: Field containing name of funder

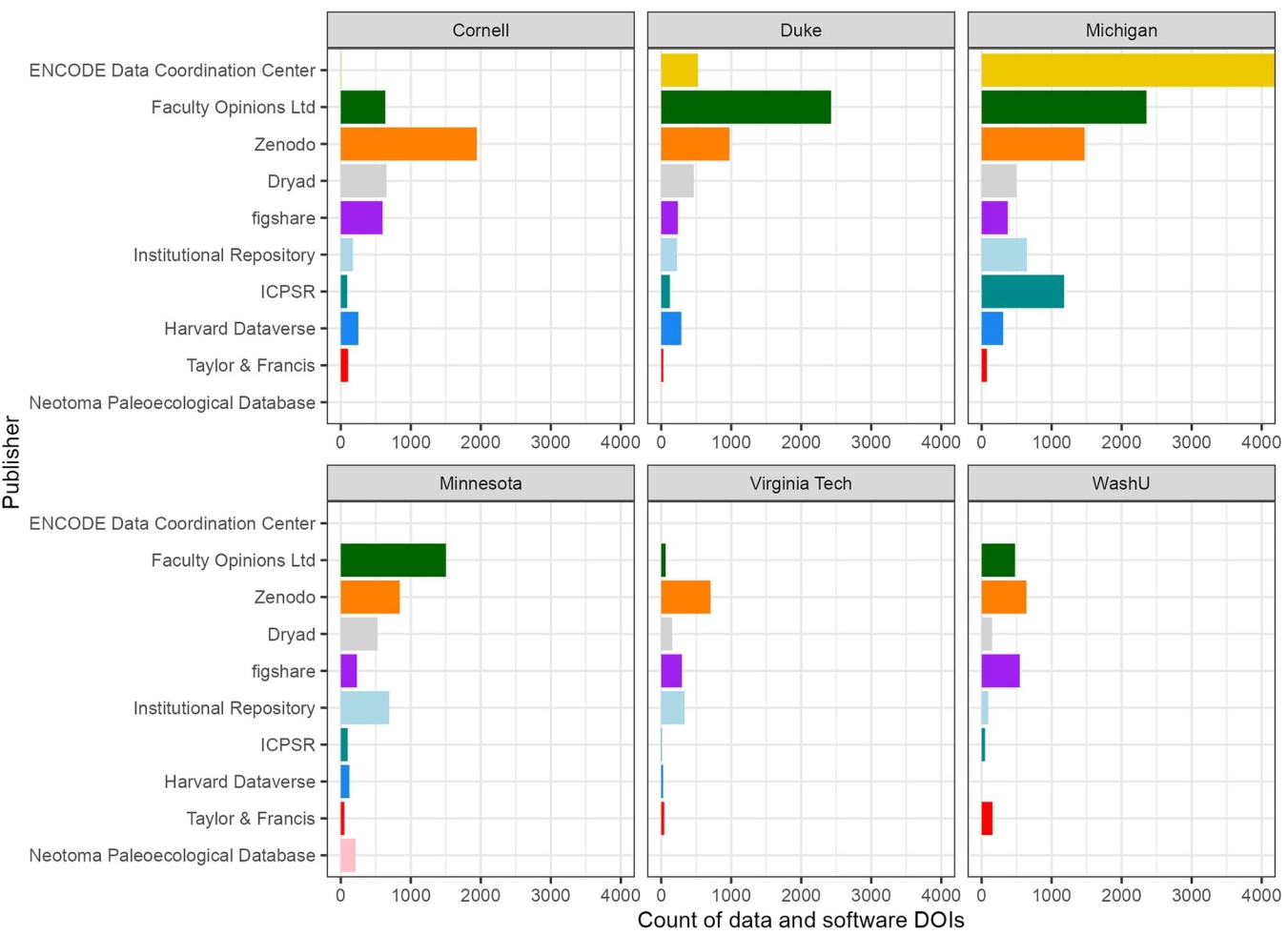

**Fig 3. DOIs in top 10 publishers by institution.** Total counts of software and data DOIs present in each of the top 10 repositories are presented by institution. The ENCODE bar at the University of Michigan is truncated at 4,000 (total N = 117,569) for the visualization to retain consistent scaling across institutions.

- Funder Award Number: Field containing award number

- Funder Identifier: PID of funding agency often a Open Funder Registry (OFR, formerly FundRef)

This preliminary step allowed us to measure the presence of information in fields likely to be at the forefront of what both funders and institutions will want to be able to query and assess as public access requirements are implemented. Here, we only assess the completeness of these fields, rather than assessing the quality of the present information. We postulate completeness is a quality measure of the record, but acknowledge that it is one piece of the quality, and that judging the quality in its entirety requires further study. The results of the metadata completeness analysis are visualized in Fig 5 and Table 5.

From our analysis, one of the common takeaways is that there are varying amounts of metadata captured by data publishers that eventually are pushed to DataCite. Researchers may simply not be providing this information, or the publisher may not be requesting this information. However, our own experiences managing local institutional repositories tells us the opposite;

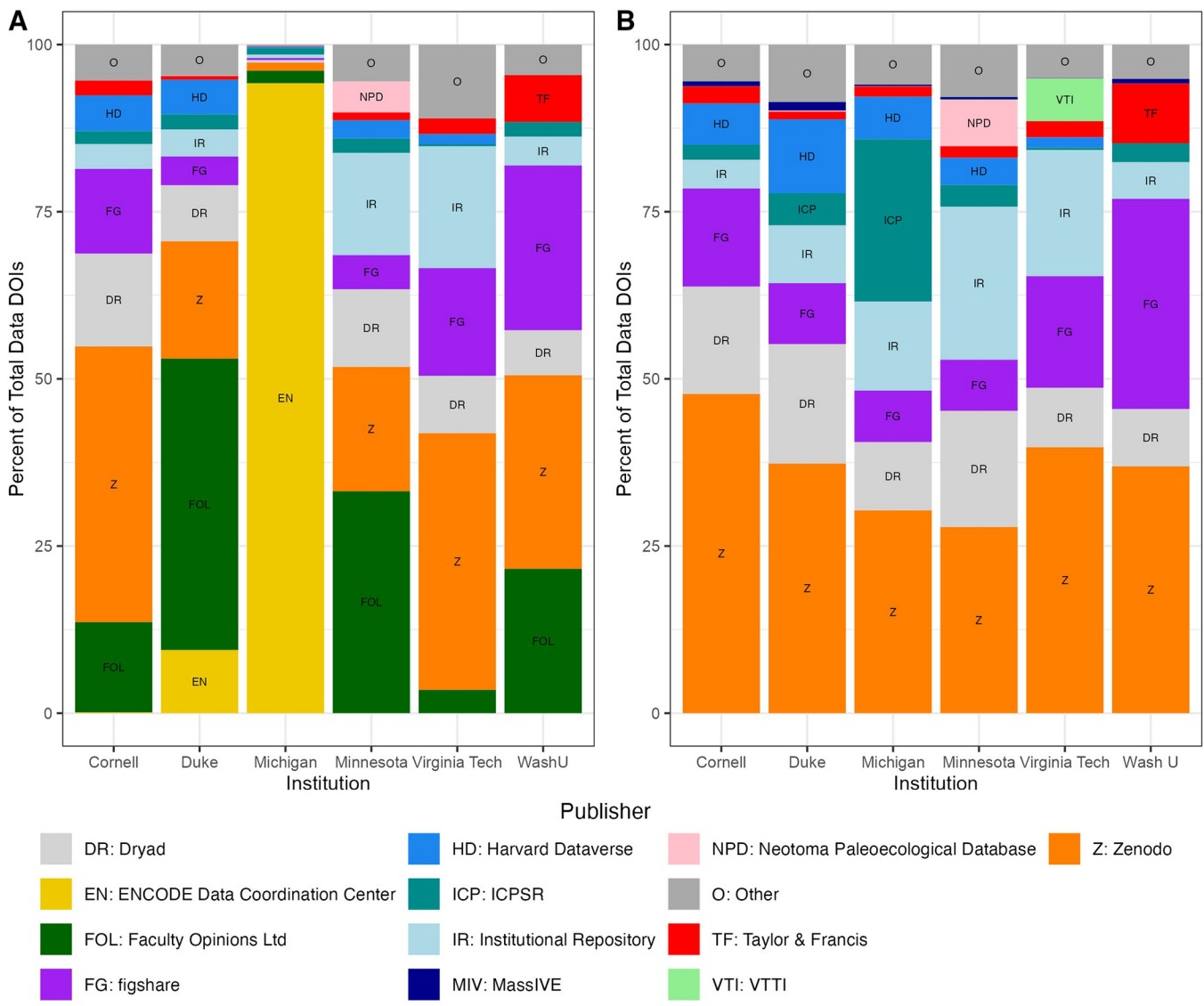

**Fig 4. Distribution of top publisher DOIs.** The data publishers are shown as a percentage of total data DOIs for each institution, with ENCODE and Faculty Opinions Ltd (A, left) and without (B, right) to show issues of scale.

that much more metadata were captured and are managed locally, but are not being entered into the DataCite record for a variety of reasons [31]. The repository may not choose to mirror their metadata (beyond the minimum required by DataCite to mint a DOI) for authority control purposes—e.g., when a change is needed, only maintaining the authoritative version in one place, or they may be in fact only be capturing minimal metadata and this is reflected in the DataCite record. As mentioned earlier, the DataCite schema has evolved over the years. The "Resource Author Affiliation" field was not introduced until DataCite Metadata Schema 3.1 [7] and RORs were not adopted into the schema until DataCite Metadata Schema 4.3 [10].

## Discussion

Overall, we found that researchers at our institutions are sharing data in an increasingly diverse number of repositories, with generalist data repositories and a few large-scale

**Table 3. Top 20 publishers of datasets and software code by affiliation.**

| rank | Publisher | Cornell University | Duke University | University of Michigan | University of Minnesota | Virginia Tech | WashU | Total | Cum. % |
|---|---|---|---|---|---|---|---|---|---|
| 1 | ENCODE Data Coordination Center | 6 | 526 | 117569 | 0 | 0 | 0 | 118101 | 82.2 |
| 2 | Faculty Opinions Ltd | 637 | 2426 | 2355 | 1503 | 64 | 478 | 7463 | 87.4 |
| 3 | Zenodo | 1944 | 975 | 1471 | 842 | 706 | 640 | 6578 | 92.0 |
| 4 | Dryad | 655 | 467 | 496 | 526 | 158 | 149 | 2451 | 93.7 |
| 5 | figshare | 597 | 238 | 373 | 231 | 296 | 545 | 2280 | 95.3 |
| 6 | Institutional Repository | 175 | 226 | 646 | 692 | 335 | 95 | 2169 | 96.8 |
| 7 | ICPSR | 92 | 125 | 1178 | 99 | 6 | 49 | 1549 | 97.9 |
| 8 | Harvard Dataverse | 251 | 289 | 308 | 123 | 28 | 0 | 999 | 98.6 |
| 9 | Taylor & Francis | 105 | 30 | 75 | 52 | 43 | 155 | 460 | 98.9 |
| 10 | Neotoma Paleoecological Database | 0 | 4 | 0 | 210 | 0 | 0 | 214 | 99.0 |
| 11 | VTTI | 0 | 0 | 1 | 2 | 113 | 0 | 116 | 99.1 |
| 12 | MassIVE | 30 | 34 | 12 | 11 | 1 | 12 | 100 | 99.2 |
| 13 | SciELO journals | 22 | 12 | 12 | 6 | 10 | 12 | 74 | 99.2 |
| 14 | Code Ocean | 11 | 17 | 20 | 16 | 0 | 0 | 64 | 99.3 |
| 15 | Borealis | 11 | 15 | 9 | 20 | 5 | 0 | 60 | 99.3 |
| 16 | OBIS-SEAMAP | 0 | 49 | 0 | 0 | 0 | 0 | 49 | 99.4 |
| 17 | Future Science Group | 4 | 2 | 0 | 4 | 22 | 12 | 44 | 99.4 |
| 18 | Authorea, Inc. | 6 | 5 | 10 | 7 | 0 | 12 | 40 | 99.4 |
| 19 | SAGE Journals | 2 | 2 | 12 | 2 | 0 | 18 | 36 | 99.5 |
| 20 | KNB Data Repository | 19 | 10 | 2 | 2 | 1 | 0 | 34 | 99.5 |
| Total by Institution | | 4,716 | 5,564 | 124,774 | 4,528 | 1,839 | 2,212 | 143,633 | 100 |
| Percent of total in Top 10 Publishers | | 94.6% | 95.4% | 99.8% | 94.5% | 89.0% | 95.4% | 99% | |

disciplinary repositories among the most popular locations to share data. We found a proliferation of repositories in which data DOIs are shared over the ten year time span we searched, with the number of repositories nearly tripling within each of our institutions. This could reflect the widening use of DOIs and associated metadata over this time, resulting in our search results returning more existing repositories across time, as well as a general increase in the breadth of data sharing accompanying federal policy changes [32, 33] and increased attention to scientific reproducibility [34, 35].

Two Crossref publishers (ENCODE data coordination center and Faculty Opinions LTD) were among the top repositories in our results, in terms of the number of issued data DOIs.

**Table 4. Elements for DataCite Metadata Schema.**

| Mandatory Properties | Recommended Properties | Optional Properties |
|---|---|---|
| *DOI** | Subjects | Language |
| State Selection | Contributors | Alternative Identifiers |
| URL | *Affiliation** | Rights |
| *Creators** | Dates | Sizes |
| Title | *Related Identifiers** | Formats |
| Publisher | Descriptions | Version |
| *Publication Year** | Geolocations | *Funding References** |
| Resource Type General | | Related Items |

Elements for DataCite Metadata Schema 4.4 [30], released 30 Mar 2021

*Metadata fields noted in the 2022 OSTP memo.

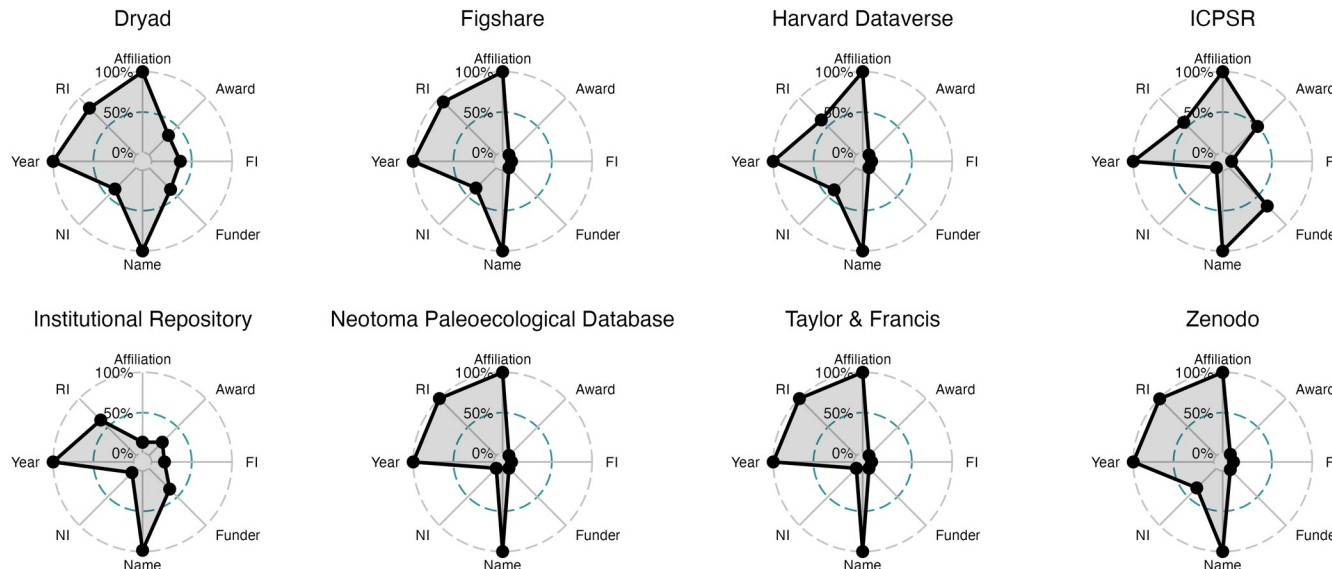

**Fig 5. Completeness of metadata fields for top DataCite repositories.** Percent of DOIs with complete (non-blank or missing) metadata for fields recommended in the 2022 OSTP memo. Year: Publication Year; Name: Creator Name; Affiliation: Creator Affiliation; RI: Related Identifiers; NI: Creator Name Identifier; Funder: Funder Name; Award: Funder Award Number; FI: Funder Identifier.

ENCODE was close to two orders of magnitude above the next highest data repository. While ENCODE accounted for 82% of the total numbers of data DOIs in our search (and 94% of the University of Michigan's results), we could not find clear ways DOIs were collapsible or able to be grouped as with other repositories that assigned DOIs per file, rather than per study. Instead, the large number of ENCODE DOIs likely reflects the nature of genetic data—which is gathered and analyzed in large quantities and at high volumes. Another top repository in our results, Faculty Opinions LTD, was found to be a journal of article reviews that used the "dataset" attribute in Crossref because of the lack of better terminology that fit that content. The other publishers were a combination of popular generalist repositories (Zenodo, figshare,

**Table 5. Metadata fields noted in the 2022 OSTP memo, by publisher and by percent complete.**

| DataCite Metadata Field | Dryad | Harvard Dataverse | ICPSR | Institutional Repository | Neotoma Paleoecological Database | Taylor & Francis | Zenodo | figshare |
|---|---|---|---|---|---|---|---|---|
| Publication Year | 2451 (100%) | 999 (100%) | 1549 (100%) | 1943 (100%) | 214 (100%) | 460 (100%) | 6578 (100%) | 2280 (100%) |
| Creator Name | 2451 (100%) | 999 (100%) | 1549 (100%) | 1914 (98.5%) | 214 (100%) | 460 (100%) | 6578 (100%) | 2280 (100%) |
| Creator Affiliation | 2451 (100%) | 999 (100%) | 1549 (100%) | 259 (13.3%) | 214 (100%) | 460 (100%) | 6578 (100%) | 2280 (100%) |
| Related Identifiers | 2016 (82.3%) | 615 (61.6%) | 888 (57.3%) | 1212 (62.4%) | 214 (100%) | 460 (100%) | 6550 (99.6%) | 2121 (93%) |
| Creator Name Identifier | 917 (37.4%) | 389 (38.9%) | 0 (0%) | 146 (7.5%) | 0 (0%) | 0 (0%) | 2266 (34.4%) | 815 (35.7%) |
| Funder Name | 933 (38.1%) | 0 (0%) | 1041 (67.2%) | 710 (36.5%) | 0 (0%) | 0 (0%) | 150 (2.3%) | 2 (0.1%) |
| Funder Award Number | 844 (34.4%) | 0 (0%) | 776 (50.1%) | 458 (23.6%) | 0 (0%) | 0 (0%) | 150 (2.3%) | 2 (0.1%) |
| Funder Identifier | 878 (35.8%) | 0 (0%) | 0 (0%) | 313 (16.1%) | 0 (0%) | 0 (0%) | 150 (2.3%) | 0 (0%) |

Dryad, Harvard Dataverse), larger scale disciplinary repositories (ICPSR, Neotoma Paleo-ecological Database), and journal-partnered repositories (Taylor and Francis), in addition to the institutional repositories.

However, these findings are importantly caveated by variations in the completeness of the DOI metadata record. As we saw early on in our study, retrieving data DOIs from our institutional researchers requires the DOI metadata to include affiliation information. This is not a required field in either the DataCite or Crossref schemas, and one that we do not consistently add to the metadata associated with DOIs generated by our own institutional repositories. As such, these repositories, which are institutionally affiliated by design, were initially absent from our affiliation search. While we adjusted our methods to retrieve DOIs for our institutional repositories by publisher name, we recognize that incomplete affiliation fields in the DOI metadata record likely excluded many repositories in which affiliated researchers share their data and code. Other studies have also documented that affiliation metadata fields are often incomplete [36, 37]. This update to our methods raised the count of datasets published by our institutional repositories in our analysis, moving these repositories as a group up to the sixth highest publisher. If we had not adjusted our methods, institutional repositories would still have been within the top 10, falling just ahead of the Neotoma Paleoecological Database with 226 datasets. It is likely the rankings of other repositories would shift if there were similar alternative ways to identify institutional affiliation within the databases.

While the generalist and large-scale repositories at the top of our publisher list do ingest large numbers of studies, it is also possible their scale and established infrastructure better position them to automate the pulling or completion of metadata fields like affiliation from the repository record to the DOI record. While many repositories collect more information than is captured in the DOI metadata record, transferring this documentation into standardized, machine-actionable formats and ontologies is a non-trivial undertaking that likely requires both human and programmatic effort. While some metadata fields may be collected in a standardized and easily transferred format by programmers, other information is often nested in files or other non-standardized formats and would require considerable human effort to clean and transfer. We also found cases where a repository's infrastructure for generating DOIs lead to the systematic exclusion of them from our search criteria. For example, the Open Science Framework (OSF) is a widely used repository platform in the social sciences and one we were surprised did not appear in our search results. Upon investigation, we found that the automated DataCite DOIs minted by that repository were associated with a "text" resource type and only included institutional affiliation for researchers with OSF Institutional Membership (see Table 6 "Where's the Open Science Framework in our analysis?").

However, when examining the completeness of other fields important for the sharing of federally funded data, as recommended in the 2022 OSTP memo, we found that even the large-scale generalist repositories do not capture all the fields for the majority of their datasets.

**Table 6. Where's the open science framework in our analysis?.**

As an example of how hard it is to capture a complete record of where data is published for a particular institution, we investigated one generalist repository that is known to be commonly used amongst our researchers, but was not represented in our sample—the Open Science Framework (OSF).

• **Affiliation Metadata:** Although researchers from all six institutions use the OSF to publish data, our search by institution only returned DOIs from the three institutions that had institutional OSF accounts (at the time of search, Virginia Tech, Cornell University, and Duke University). This account type allowed affiliation to be automatically pulled into the DOI metadata.

• **Resource Type Metadata:** The OSF is a platform that houses documentation as well as data, code, and other study materials. Of the 1,187 DOIs we initially found from the institutions with affiliation metadata, all of these were labeled as "text" and did not meet our inclusion criteria of having a resource type of "dataset" or "software."

While not all shared data are generated from funded research (and thus would not have applicable funder metadata), other fields were also absent from the majority of DOIs that *are* applicable to all submissions, such as creator name identifiers (e.g., ORCIDs). It is notable that ICPSR stood out among the top DataCite repositories in terms of the presence of funder identifiers, such as the name and award number. This may be a result of ICPSR's partnership with federal agencies such as the National Institute on Drug Abuse (NIDA) [38], and hosted associated repositories, like the National Addiction and HIV Data Archive Program (NAHDAP) [39]. Therefore, ICPSR may house a greater number of datasets produced by federally funded research and more deliberately ensure that funder metadata is captured and incorporated into the DOI metadata record through established curation procedures.

One potentially concerning take away from this study is the high rates of incomplete metadata within the DOI records. Although this significantly limits DOI-based searches like ours, it is important to note that the metadata present in the DataCite and Crossref records are not fully reflective of the metadata associated with the data files or studies. DataCite requires a minimal set of mandatory metadata fields (see the fields in Table 4) to issue a DOI, resulting in only a few fields that have consistently populated fields across all generated DOIs. Depending on repository infrastructure and set up for DOI issuing, some fields may be automatically pulled into the record. Other fields may need to be manually entered. Harvesting metadata using the Open Archives Initiative Protocol for Metadata Harvesting (OAI-PMH) protocol, for example, retrieves more metadata than those fields required by DataCite but is still limited to the 15 Dublin Core elements set without qualification [40], which does not include author or creator affiliation [41]. Institutional and other data repositories often contain additional metadata that may be missed if they fall outside of the boundaries of the protocols used for harvesting. There is also a potential wealth of descriptive and other information contained in ReadMe files or other documentation that typically accompanies a dataset, and that could serve as valuable metadata if it were repurposed and made more structured [31]. Restructuring documentation to a machine readable format could not only enable more inclusive search results, but provide more accurate accounts of where and how data are shared, and connect locally-stored metadata into the broader research infrastructure. However, this is not without investment in terms of time, effort, and infrastructure.

Even within metadata fields that were complete, we also observed lack of consensus and clarity in how fields were used or defined by repositories and publishers. For example, the use of "dataset" in the resource type field was used by at least one publisher for review articles, but also covered both data and code associated with a study. While we included the term "software" for DataCite repositories, it was clear this label was used much less frequently than "dataset", since some DOIs contained both numeric data and statistical code files. Moreover, some repositories issued DOIs to each file, while others assigned DOIs at the study level. Despite our efforts to clean the data to some extent by collapsing files based on container IDs, not all repositories used this field. Furthermore, multiple versioned datasets associated with multiple DOIs were not removed or accounted for, as there did not appear to be a consistent way publishers indicated version information. This wide interpretation of metadata fields and how they were populated made it difficult to feel confident that we were truly comparing apples to apples in the metadata counts and assessment. The granularity to which an identifier should be given is a pressing question, and is the topic of continuing discussion [42].

Our study used metadata available only through DataCite and Crossref. The APIs available with DataCite and Crossref allow for systematic analysis and interpretation of their available metadata about datasets generated at our six institutions, and such analysis and interpretation is crucial for a broad view of the data sharing landscape. However, from the related studies in this RADS project (15) we know there are many other datasets from our six institutions

available via repositories and other online interfaces that do not send separate dataset metadata to either DataCite and Crossref, such as in supplementary materials, personal websites, or repositories that issue non-DOI persistent links. The scope, number, and associated metadata of these other datasets have been omitted from this analysis. This is not a trivial gap. For example, one recent study found only a third of their institution's published datasets were shared with a DOI [12].

However, as curators and data professionals within our institutions, we see these findings as an opportunity to reflect on our own processes and procedures around metadata. How can the data community, DOI minting authorities, and data repositories work to improve completeness and quality of the metadata across global infrastructures such as DataCite and Crossref? In our recommendations, we describe actions that can be taken by each of these groups to improve both the findability and quality data sharing.

## Recommendations

We recommend these concrete steps that repositories and data authors can take to improve the metadata so datasets and research outputs are more contextually connected and analysis such as this might be more successful in the future.

1. The data community should:

○ Strive for a shared protocol for data description to aid in search and retrieval. In the current U.S. environment, metadata aggregation for shared/published datasets includes disparate sources: DataCite is one source, Crossref is another. Each source requires a different metadata profile and, hence, different standards for metadata completeness. Other more specific federated systems for research data metadata, such as the European Open Science Cloud (EOSC), have aggregated and provide a unified search interface across sources despite metadata challenges. As the Open Science movement progresses, we need a shared standard for searching across repository sources, such as the Open Archives Initiative Protocol for Metadata Harvesting (OAI-PMH), or data models such as OpenAIRE Graph. The community should advocate for this marketplace and data contributors, such as Google Dataset Search which recommends repositories use schema.org structures to elevate their metadata about shared data [43], need to be transparent instead of using "black box" techniques with difficult to interpret sources and opaque data cleaning techniques [44].

○ Establish consistent guidance for how DOIs are applied and referenced within components of a study (e.g., across data, code, procedures, etc). The level of granularity for a data file vs. a dataset vs. a data collection is a needed area of clarification, so that we might compare data repository holdings more accurately. Additionally, the reference to a DOI (e.g., cited by, supplement to) should be more accurately defined and standardized to better clarify the relationships between objects. Doing this will help better establish meaningful linkages between different components of a research project, allowing users and institutions to find and explore research more holistically through data models and knowledge graphs.

○ Advocate to funders and other policy makers to help track and make better connections with published datasets and other research materials as a requirement for grant recipients. An example of this could be providing persistent identifiers for a grant in order to be able to connect outputs of that grant to one another, such as the Award DOI Service [45].

2. DOI minting authorities should:

○ Collectively determine a shared controlled vocabulary for asset type and establish guidelines for their application. For example: when should an asset be labeled a "dataset" and when it

is not a dataset? Is "code" the same as "software"? DOI registration agencies should also prioritize consistency across shared vocabularies used. For instance, DataCite has the option to distinguish between "software" and "dataset" within resource types, while Crossref does not. We should agree as a community how DOIs are consistently applied, and for all types of research outputs.

3. Data repositories should:

○ Prioritize data description and curation. As noted earlier, both the literature and scholarly communications community provide ample advice, best practices, and standards focusing on required practices for data description and sharing that are necessary to advance data discovery and reuse. Institutions seeking to improve data description and sharing must translate *best* practices, which may be too high of a bar to actually implement, to *effective* practices at their institution, taking into account institutional goals and locally supported resources.

○ Ensure that author affiliation metadata is a *required* field for dataset publication. RORs and ORCIDs are standard practices for collecting authorship affiliation, and must be collected and associated with each author in a dataset record [46].

○ Implement data sharing practices and infrastructure decisions with the pipeline to the global metadata infrastructure in mind. For example, requiring collection of federally recommended metadata fields [47] in machine actionable formats to allow automated porting and crosswalking to DOI infrastructure.

## Conclusion

The process of seeking and both finding and not finding data shared by researchers affiliated with six large US research universities presented us with several challenges. While we found data across a number of generalist and disciplinary repositories, we are certain there are many datasets we were unable to find. It is important to understand where researchers are sharing their data to meet open data requirements and goals as this information helps universities, researchers, and libraries determine the criticality of this infrastructure, and where to invest planning and sustaining efforts going forward. The metadata quality landscape for published research datasets is uneven and key information, such as author affiliation, is often incomplete or missing from source data repositories and aggregators.

Understanding the impact of data sharing and related open science activities is hindered by the lack of available and detailed metadata about the research being shared. With the results and the recommendations shown here, we aim to move our community forward and better serve the research community.

## Acknowledgments

We would especially like to thank Martin Halbert, Program Director at the National Science Foundation, for his considerations regarding interoperable research infrastructure, data, and metadata. We would also like to thank Ted Habermann, of Metadata Game Changers, for his work on the RADS study and contributions to this research, and to DataCite and Crossref for providing their APIs. We would also like to thank the members of the Data Curation Network for engagement and discussion around these topics. Finally, we thank Mikala Narlock, Megan O'Donnell, Sara Mannheimer, and Kristin Briney for providing early feedback on a draft of this paper.

## Author Contributions

**Conceptualization:** Lisa R. Johnston, Joel Herndon, Shawna Taylor, Jake R. Carlson, Jennifer Moore, Jonathan Petters, Wendy Kozlowski, Cynthia Hudson Vitale.

**Data curation:** Alicia Hofelich Mohr.

**Formal analysis:** Alicia Hofelich Mohr.

**Investigation:** Lisa R. Johnston, Alicia Hofelich Mohr, Jake R. Carlson.

**Methodology:** Lisa R. Johnston, Alicia Hofelich Mohr.

**Project administration:** Joel Herndon, Shawna Taylor, Cynthia Hudson Vitale.

**Software:** Alicia Hofelich Mohr.

**Validation:** Lisa R. Johnston, Lizhao Ge.

**Visualization:** Lisa R. Johnston, Alicia Hofelich Mohr.

**Writing – original draft:** Lisa R. Johnston, Alicia Hofelich Mohr, Joel Herndon, Shawna Taylor, Cynthia Hudson Vitale.

**Writing – review & editing:** Lisa R. Johnston, Alicia Hofelich Mohr, Joel Herndon, Shawna Taylor, Jake R. Carlson, Lizhao Ge, Jennifer Moore, Jonathan Petters, Wendy Kozlowski, Cynthia Hudson Vitale.

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
