## [Decision Letter · Decision Letter 0]

14 Feb 2024

PONE-D-23-35578Seek and you may (not) find: A multi-institutional analysis of where research data are sharedPLOS ONE

Dear Dr. Herndon,

Thank you for submitting your manuscript to PLOS ONE. After careful consideration, we feel that it has merit but does not fully meet PLOS ONE’s publication criteria as it currently stands. Therefore, we invite you to submit a revised version of the manuscript that addresses the points raised during the review process.

The study discussed here explores how academic research data is shared, focusing on understanding the processes and expenses involved in making this data accessible to the public. The research takes a close look at where scholars commonly share their data. The article is intriguing because it takes a practical approach to tackling the problem. However, it suggests that there's room for improvement. For instance, the methods used in the study could be better explained to give readers a clearer understanding of how the research was conducted. Additionally, the findings could be expanded upon to provide more detailed insights. By refining these aspects, the study could become even more valuable in shedding light on scholarly publishing practices and the accessibility of research data.

We look forward to receiving your revised manuscript.

Kind regards,

Ayesha Maqbool, PhD

Academic Editor

PLOS ONE

 [CHV

NSF award #2135874

National Science Foundation

https://www.nsf.gov/

No].  

[Funding for this research and the Realities of Academic Data Sharing (RADS) Initiative was provided by the National Science Foundation (NSF), award #2135874, EAGER grant: Completing the Lifecycle: Developing Evidence Based Models of Research Data Sharing. We would especially like to thank Martin Halbert, Program Director at NSF, for his considerations regarding interoperable research infrastructure, data, and metadata. We would also like to thank Ted Habermann, of Metadata Game Changers, for his work on the RADS study and contributions to this research, and to DataCite and Crossref for providing their APIs. We would also like to thank the members of the Data Curation Network (DCN) for engagement and discussion around these topics. Finally, we thank Mikala Narlock, Megan O’Donnell, Sara Mannheimer, and Kristin Briney for providing early feedback on a draft of this paper.]

  [CHV

NSF award #2135874

National Science Foundation

https://www.nsf.gov/

No].  

[I have read the journal's policy and the authors of this manuscript have the following competing interests: 

Institutions involved in this study maintain paid memberships in either the CrossRef or DataCite data sharing services.]. 

Reviewers' comments:

Reviewer's Responses to Questions

**Comments to the Author**

1. Is the manuscript technically sound, and do the data support the conclusions?

Reviewer #1: Partly

Reviewer #2: Yes

Reviewer #3: Yes

2. Has the statistical analysis been performed appropriately and rigorously? 

Reviewer #1: Yes

Reviewer #2: Yes

Reviewer #3: N/A

3. Have the authors made all data underlying the findings in their manuscript fully available?

Reviewer #1: Yes

Reviewer #2: Yes

Reviewer #3: Yes

4. Is the manuscript presented in an intelligible fashion and written in standard English?

Reviewer #1: Yes

Reviewer #2: Yes

Reviewer #3: Yes

5. Review Comments to the Author

Reviewer #1: This paper focuses on answering three research questions in order to shed light on (i) where research data is shared, (ii) how many datasets are shared, as well as (iii) completeness of datasets' metadata.

Research conducted in this paper is well motivated and it seems very relevant to me, since it allows reader to understand how different stakeholders (not only researchers) are engaged with open science. Other strong point of the paper is that a method is proposed for other researchers and academic institutions to better know how data sharing practices are being applied (beyond the ones considered in this paper). Actually, other strength of this paper is that authors consider a bottom-up approach, thus being a realistic overview of current application of open science practices.

However, the paper has some weak points, authors need to consider in a revised manuscript:

1-Chart in figure 5 (a line chart) is not the best way to visualize percentage data. I would recommend authors to consider other kind of charts that better fits with percentage data. I would recommend authors to use radar charts, but they may want to consider some kind of pie charts.

2-Metadata that is considered is not explicitly described in detail. There are some concepts that are crystal clear, such as DOI, but others need to be further described, such as "Related Identifiers". Please, provide a more detailed explanation of the considered metadata (highlighted concepts in Table 4).

3-Authors provide good recommendations but I would suggest authors to create a specific section for recommendations (and not mixing conclusions with recommendations in a unique section). Also, this new recommendation section must be more long with more complete explanations.

4-Research questions that are stated at the beginning of the paper must be explicitly answered when data has been analyzed.

5-Authors should consider and summarize other related work in order to explicitly state their contributions with regard to related work. Also, you may consider other related initiatives around the world, such as EOSC (https://eosc-portal.eu/).

Reviewer #2: This is an interesting paper about using repositories for sharing of research data based on a bibliographic analysis. It provides data on which repositories are used for data sharing by researchers from 6 US institutions and about the completeness of metadata fields for top DataCite repositories. The paper is well written but needs some improvement:

Methods:

From the viewpoint of the reviewer, the presentation of the methods could be made clearer. It should be stated whether a study protocol was available before start of the study and whether such a protocol was registered or not. It should also be made clear whether the addition of a 3rd group changed after initial analysis (as it seems to be) and whether that resulted in an updated study protocol.

The section “data collection – institutional repositories” is difficult to understand. There is a mix of datasets related to the listed institutional repositories (2164, see table 1) and a remaining number of institutional repositories (2390 minus 2164, table 2) running through a cleaning/curating process. If the reviewer understood it correctly, for the institutional dataset, no cleaning/curating was necessary because of the pre-selection of this dataset according to the defined and listed repositories.

Taken into consideration that a different search strategy was used for the DataCite/Crossref and the institutional sample, it could be of advantage to separate the analysis and not to merge all the data into one sample.

In figure 1 filtering according to resource type = dataset is missing for Institutional Repositories (IRs) and Crossref.

Results

In table 3 the institutional repositories are ranked no. 5. As explored, the datasets here are mainly from the listed institutional repositories with some additional datasets retrieved from Datacite and Crossref search. Because a different search strategy was used for the DataCite/Crossref and the institutional sample, it would be clearer, to separate this row into 2 subgroups.

Discussion

The discussion about the unit of analysis and the consequences of it (e.g. DOI per file, DOI per study) is important. There is still no common and standardised definition of a research study or a research project, to which a dataset can be linked as a research output. This issue is discussed, for example, in the section “Data analysis”, referring to individual studies that may have many DOIs and where datasets falling into the same container were collapsed. This point is very relevant for any bibliographic analysis and needs more discussion from the viewpoint of the reviewer. One approach is the OpenAIRE Knowledge Graph, including metadata and links between scientific products (e.g. literature, datasets, software, and "oth-er research outputs"), organizations, funders, funding streams, projects, communities, and data sources, another approach is trying to provide a proposal for a metadata framework for contextual metadata defining research projects and linking it to research outputs (e.g. da-tasets), taking into consideration that relationships between projects/studies with fund-ing/grants and research outputs (e.g. datasets) can be difficult.

The reviewer could imagine that comparing the completeness of the DOI metadata of the institutional sample with the metadata available in the institutional repositories could be of major interest and worth to be investigated in the future. If it turns out that the metadata completeness in the institutional metadata is much higher than the DOI metadata, it is pri-marily a transfer and not documentation problem.

The reviewer agrees with most of the conclusions and recommendations, especially with respect to guidance how DOIs are applied and referenced within components of a study. This is truly an area with need for clarification. But this needs a broader discussion involving frameworks for representing research projects and research outputs, ontolo-gies/terminologies, including crosswalks and requirements for machine-actionable metadata (relationships between entities/records, unique identifiers, research graphs).

Reviewer #3: This paper is a study of metadata quality on a subset of research data. The results are not surprising (metadata are incomplete) but it is still good to put numbers on it. Since these results are limited to the six universities in question, some information about them would help determine just how representative (or not) these results are of the entirety of the academic research enterprise - something similar to a summary of demographics of study subjects. Maybe budget, size of faculty, grant dollars, etc. My guess is that some of these results are not very generalizable, which is also an interesting result.

I like the recommendations at the end.

Line 333: There is a spurious "of"

6. PLOS authors have the option to publish the peer review history of their article (what does this mean?). If published, this will include your full peer review and any attached files.

Reviewer #1: No

Reviewer #2: No

Reviewer #3: **Yes: **Anne E Thessen

---

## [Author Response · Author response to Decision Letter 0]

29 Mar 2024

Detailed responses to each reviewer comment are provided below (denoted by a 'Response:').

Reviewer's comments to authors

Reviewer #1

 This paper focuses on answering three research questions in order to shed light on (i) where research data is shared, (ii) how many datasets are shared, as well as (iii) completeness of datasets' metadata.

Research conducted in this paper is well motivated and it seems very relevant to me, since it allows reader to understand how different stakeholders (not only researchers) are engaged with open science. Other strong point of the paper is that a method is proposed for other researchers and academic institutions to better know how data sharing practices are being applied (beyond the ones considered in this paper). Actually, other strength of this paper is that authors consider a bottom-up approach, thus being a realistic overview of current application of open science practices.

However, the paper has some weak points, authors need to consider in a revised manuscript:

1-Chart in figure 5 (a line chart) is not the best way to visualize percentage data. I would recommend authors to consider other kind of charts that better fits with percentage data. I would recommend authors to use radar charts, but they may want to consider some kind of pie charts.

Response: Thank you for this suggestion. We have updated figure 5 to use a radar chart that better fits the percentage data. 

2-Metadata that is considered is not explicitly described in detail. There are some concepts that are crystal clear, such as DOI, but others need to be further described, such as "Related Identifiers". Please, provide a more detailed explanation of the considered metadata (highlighted concepts in Table 4).

Response: Thank you. We have defined the metadata elements used in the analysis. 

3-Authors provide good recommendations but I would suggest authors to create a specific section for recommendations (and not mixing conclusions with recommendations in a unique section). Also, this new recommendation section must be more long with more complete explanations.

Response: Thank you. We have separated out the recommendations from the conclusions with additional details added. 

4-Research questions that are stated at the beginning of the paper must be explicitly answered when data has been analyzed.

Response: We have reformatted the results section to reflect the answers to the research questions. 

5-Authors should consider and summarize other related work in order to explicitly state their contributions about related work. Also, you may consider other related initiatives around the world, such as EOSC (https://eosc-portal.eu/).

Response: We had not considered this project in relation to our work, however, thanks for making us aware and we will investigate in future. We appreciate the reviewer’s suggestion on other models and infrastructure for finding and discovering research data shared internationally, and we have added reference to it in our recommendations. 

Reviewer #2: 

This is an interesting paper about using repositories for sharing of research data based on a bibliographic analysis. It provides data on which repositories are used for data sharing by researchers from 6 US institutions and about the completeness of metadata fields for top DataCite repositories. The paper is well written but needs some improvement:

Methods:

From the viewpoint of the reviewer, the presentation of the methods could be made clearer. It should be stated whether a study protocol was available before the start of the study and whether such a protocol was registered or not. It should also be made clear whether the addition of a 3rd group changed after initial analysis (as it seems to be) and whether that resulted in an updated study protocol.

Response: Thank you for pointing out this issue. We did not register our protocol for this study since it was exploratory in nature. We have noted this in the Methods section.

The section “data collection – institutional repositories” is difficult to understand. There is a mix of datasets related to the listed institutional repositories (2164, see table 1) and a remaining number of institutional repositories (2390 minus 2164, table 2) running through a cleaning/curating process. If the reviewer understood it correctly, for the institutional dataset, no cleaning/curating was necessary because of the pre-selection of this dataset according to the defined and listed repositories.

Response: We have rearranged this section and included the counts of institutional repositories found in the original affiliation based searches to help make this method clearer. We also thank this reviewer for helping us identify that the results had not been sufficiently de-duplicated. We have made that change throughout the paper. 

Taken into consideration that a different search strategy was used for the DataCite/Crossref and the institutional sample, it could be of advantage to separate the analysis and not to merge all the data into one sample.

Response: Thank you for these suggestions. Different search strategies were necessary given the variability of the metadata fields found in the DataCite/Crossref API’s. While the search methods differed slightly, the results of this study are ultimately unchanged by pulling out these results separately. We have added a few sentences to the paper discussing the results of the IR search prior to the merge in order to more clearly show the results differences. 

In figure 1 filtering according to resource type = dataset is missing for Institutional Repositories (IRs) and Crossref.

Response: Thank you, this was unintentionally omitted. We have corrected this in the figure.

In table 3 the institutional repositories are ranked no. 5. As explored, the datasets here are mainly from the listed institutional repositories with some additional datasets retrieved from Datacite and Crossref search. Because a different search strategy was used for the DataCite/Crossref and the institutional sample, it would be clearer, to separate this row into 2 subgroups.

Response: Thank you. We have presented the results for IRs without the specific search to help address how the change in method affected the results. Because this only affected the counts of IRs, we left the rest of the analyses together. We also added discussion of how the results were impacted by using this different method. 

Discussion:

The discussion about the unit of analysis and the consequences of it (e.g. DOI per file, DOI per study) is important. There is still no common and standardized definition of a research study or a research project, to which a dataset can be linked as a research output. This issue is discussed, for example, in the section “Data analysis”, referring to individual studies that may have many DOIs and where datasets falling into the same container were collapsed. This point is very relevant for any bibliographic analysis and needs more discussion from the viewpoint of the reviewer. One approach is the OpenAIRE Knowledge Graph, including metadata and links between scientific products (e.g. literature, datasets, software, and "oth-er research outputs"), organizations, funders, funding streams, projects, communities, and data sources, another approach is trying to provide a proposal for a metadata framework for contextual metadata defining research projects and linking it to research outputs (e.g. da-tasets), taking into consideration that relationships between projects/studies with fund-ing/grants and research outputs (e.g. datasets) can be difficult.

Response: We have added references to OpenAIRE and discussion of linkages across products. 

The reviewer could imagine that comparing the completeness of the DOI metadata of the institutional sample with the metadata available in the institutional repositories could be of major interest and worth to be investigated in the future. If it turns out that the metadata completeness in the institutional metadata is much higher than the DOI metadata, it is pri-marily a transfer and not documentation problem.

Response: Thank you for this comment. We agree transfer is part of the problem and added text in the discussion to address this point. 

The reviewer agrees with most of the conclusions and recommendations, especially with respect to guidance how DOIs are applied and referenced within components of a study. This is truly an area with need for clarification. But this needs a broader discussion involving frameworks for representing research projects and research outputs, ontolo-gies/terminologies, including crosswalks and requirements for machine-actionable metadata (relationships between entities/records, unique identifiers, research graphs).

Response: We appreciate the detailed analysis here and the support of our findings. We have expanded on some of these points in our recommendations. 

Reviewer #3: 

This paper is a study of metadata quality on a subset of research data. The results are not surprising (metadata are incomplete) but it is still good to put numbers on it. Since these results are limited to the six universities in question, some information about them would help determine just how representative (or not) these results are of the entirety of the academic research enterprise - something similar to a summary of demographics of study subjects. Maybe budget, size of faculty, grant dollars, etc. My guess is that some of these results are not very generalizable, which is also an interesting result.

Response: Thank you for this suggestion. We added more information about these universities in the methods section. 

I like the recommendations at the end.

Line 333: There is a spurious "of"

Response: Thank you, this has been omitted.

---

## [Editor Report · Decision Letter 1]

4 Apr 2024

Seek and you may (not) find: A multi-institutional analysis of where research data are shared

PONE-D-23-35578R1

Dear Dr. Herndon,

We’re pleased to inform you that your manuscript has been judged scientifically suitable for publication and will be formally accepted for publication once it meets all outstanding technical requirements.

Kind regards,

Ayesha Maqbool, PhD

Academic Editor

PLOS ONE